# Investigation on Microstructure, Mechanical and Wear Properties of HVOF Sprayed Composite Coatings (WC–Co + CR) On Ductile Cast Iron

**DOI:** 10.3390/ma14123282

**Published:** 2021-06-14

**Authors:** Marzanna Ksiazek, Ilona Nejman, Lukasz Boron

**Affiliations:** 1Department of Non-Ferrous Metals, AGH University of Science and Technology, 30 Mickiewicza Ave., 30-059 Cracow, Poland; inejman@agh.edu.pl; 2Łukasiewicz Research Network—Cracow Technology Institute, 73 Zakopianska St., 30-418 Cracow, Poland; lukasz.boron@kit.lukasiewicz.gov.pl

**Keywords:** WC–Co coating, thermal spraying, HVOF, scratch test, wear resistant

## Abstract

Recent work indicates that the high-velocity oxy-fuel (HVOF) thermal spraying WC–Co coatings have been used to enhance the wear resistance of various engineering components in a variety of industrial environments. In the present work, WC–Co powder, containing Cr particles in an amount of 10%, was deposited on ductile cast iron with the HVOF thermal spray coating technique. An investigation was conducted to determine the role of Cr particles in the WC–Co coating produced with the HVOF technique on microstructure, mechanical, and wear properties in a system of type: WC-Co coating/ductile cast iron. The microstructure of the HVOF-sprayed WC–Co + Cr coating was characterised by light microscopy, X-ray diffraction (XRD), scanning electron microscope (SEM), transmission electron microscope (TEM), and energy-dispersive X-ray spectroscopy (EDS). The analysis of the microstructure showed the formation of a coating with low porosity, compact structure, and good adhesion to the substrate with a typical lamellar structure composed of fine molten Cr particles and finely fragmented WC grains embedded in a Co matrix, reaching the size of nanocrystalline. The scratch test was applied for the analysis of the adhesion of coatings to the substrate. The erosion behaviour and mechanism of material removal was studied and discussed based on microstructural examinations. Moreover, the results were discussed in relation to the bending strength test, including cracks and delamination in the system of the WC–Co + Cr/ductile cast iron, as microhardness and erosion resistance of the coating. It was found that the addition of Cr particles to the WC–Co powder, which causes hardening of the binder phase is a key influence on increased mechanical and wear properties in the studied system. Additionally, due to the construction of nanostructured coatings, suitable proportion of hard and soft phases, the technique sprayed HVOF coatings have advantageous properties such as high density and good slurry erosion resistance.

## 1. Introduction

The HVOF—that is the High-Velocity Oxy-Fuel method—is still a developing technology of surface modification, leading to increased erosion and corrosion resistance, and thus extended usage time of crucial parts of machines and devices [1]. This technology makes it possible to considerably increase the durability and reliability of machine parts, reduce production costs, and save materials and energy. In particular, carbide coatings created with thermal spraying techniques spurred interest in researchers due to their high wear resistance, high-temperature corrosion resistance, as well as them meeting the most stringent conditions for demanding energy applications [2,3]. Due to the fact that, in the HVOF process, the powder particles are accelerated with high velocity, and when they are deposited onto a substrate, they form a ‘splat by splat’ coating with good mechanical bonding properties [4]. Owing to the relatively low process temperature and due to a relatively short period of charge powder occupying the gas jet, adverse phenomena, connected with changes within the phase composition of coatings, such as carbide decomposition and oxidation of metallic and carbide materials, are limited. This allows for the production of carbide coatings with special properties, such as high hardness, high abrasion and erosion resistance, high-temperature corrosion resistance, good thermal conductivity and lower porosity, as well as higher adhesion to the substrate, in comparison to coatings generated with the use of conventional plasma spraying [5,6,7]. It must be stressed that their microstructure, as well as mechanical and tribological properties, are conditioned by HVOF spraying process parameters, such as temperature, velocity, chemical, and phase composition of charge powder, resultant porosity, and residual stresses [8]. Previous studies indicate that, during the process of HVOF method spraying, in the majority of cases, the WC–Co coatings are subject to decomposition and decarbonisation, which lead to the generation of harmful phases, for example: W_2_C, W, and amorphous or nanocrystalline phase in Co–W–C base. These phases, however, do not condition the reduction in coatings’ hardness, but they do, indeed, reduce the wear resistance [9,10].

Although, many years, industrial coatings are often added to metal parts to offer an extra layer of protection from mechanical and environmental influences—improving their performance and extending their lifetime use, and new types of carbide coatings are still under development. Subject literature indicates that studies on the improvement of mechanical and wear properties of coatings are concentrated on the right selection of tungsten carbide grain size in charge powder; modification of the chemical composition of charge powder both by changing its chemical composition and doping with metallic or ceramic particles; modification of the chemical composition of binder and coating generation process parameters [11,12]. Over the past years, it has been demonstrated many times that the use of nanostructured materials for making high quality coatings provides them with high abrasion resistance, low friction coefficient (also at high temperatures), and resistance to chemical environments [13,14]. Reduction in grain size of tungsten carbide, in a cobalt matrix, to nanometres results in the limitation of plastic deformation of the matrix and increase in hardness, and, as a consequence, increase in resistance to erosive wear of a coating [15]. In turn, the change in the chemical composition of the charge powder WC–Co to the WC–Co–Cr allowed to produce coatings having very good tribological properties, such as low coefficient of friction, high abrasion resistance (even at high temperatures and in suspension), and high resistance to chemical environments [16]. In the work of Liu et al. [5] and Ozbek et al. [17], they emphasized that the hard WC particle in the coating leads to high coating hardness and higher wear resistance, while the metal binder Co–Cr supplies the necessary coating toughness and required hardness. It has been evidenced that the admixture of chromium modifies, to some extent, the distribution of WC grains, preventing decarbonisation, while, at the same time, improving the adhesive resistance of WC grains with a cobalt matrix [18,19,20]. Moreover, these coatings provide for a better distribution of contact stresses, thus making it possible to avoid serious problems connected with delamination. On the other hand, adding Al to the nanostructured WC–12Co coating produced in the HVOF process resulted in an improvement of the mechanical characteristics of a coating, as well as an increase in abrasion resistance compared to the coating produced with the same method, but from conventional powder [21]. To our knowledge, there is still lack of systematic research determining the impact of modification of the chemical composition of the starting powder on microstructural features (metallic binder and carbide grains within the coating), mechanical, and wear properties of HVOF sprayed WC–Co coating.

This study focuses on the creation of WC–Co + Cr composite coatings using the HVOF technique, as well as on the characteristics of their microstructure, mechanical properties, and wear in order to improve the operational characteristics of the responsible parts of machines and devices, made of ductile iron. Moreover, the study assessed the effect of introducing Cr particles into the WC–12Co charge powder on the relationship between the microstructure—properties of the HVOF sprayed WC–Co coating.

## 2. Materials and Methods

### 2.1. Materials

The composite coating was created with the high-velocity flame spraying of carbide powder with a composition of WC–12Co (88 wt.%WC-12 wt.%Co), and with a grain size of −45 + 15 µm, (Diamalloy 2002 Salzer-Metco, Pfattikon, Switzerland), which was doped with 10% of Cr particles of 20 µm. For the purpose of spraying the coating, the HV-50 HVOF spraying system was used in the Plasma System S.A. company (Siemianowice, Silesia, Poland), in which the fuel used for the spraying process was a mixture of kerosene and oxygen. The substrate was made of EN-GJS-500-7 ductile cast iron with a chemical composition of: 3.61% C, 2.29% Si, 0.45% Mn, 0.045% P, 0.009% S, 0.03% Cr, 0.01% Ni, 0.057% Mg, 0.75% Cu, and the rest Fe (in weight percentage), and was characterised by the following mechanical properties are given in Table 1. Substrates had dimensions of 100 mm × 15 mm × 5 mm. Before spraying, the surface of the substrate was subject to abrasive blasting treatment with loose corundum and granulation of 20 mesh to improve the mechanical adhesion of coatings. The roughness parameter of the substrate surface Ra was 30 μm. The optimized values of process parameters used for deposition of carbide coating are listed in Table 2. Mean thickness of the coating was 200 μm.

### 2.2. Coating Charcterisation

For the study of the microstructure and chemical composition of coating/substrate type systems, light microscope (LM), scanning electron microscope (SEM), and transmission electron microscope (TEM), all equipped with energy dispersive X-ray spectrometers (EDS), were used. The coating/substrate type preparations for the transmission electron microscope in the form of a thin film were obtained with the employment of ionic thinning in a special Gatan PIPS691V3.1 device (Pleasanton, USA) for low-angle thinning [22]. Research on the phase composition was conducted with an X’Pert Pro Panalytical diffractometer (Malver Panalytical Ltd., Cambridge, UK) in the angular range of 20–90°, with CuK radiation. The measurement of porosity of the carbide coating was conducted with the use of X-ray computer tomography and Pheonix Nanotom (GE Sensing & Inspection Technologies, Ciudad de Buenos Aires, Argentina) X-ray nano-CT scanner, equipped with AxioVision 3.0 software for image analysis. The research was conducted for ten areas within the coating.

Measurements of the microhardness of coatings were conducted on samples made on cross-sections of samples normal to their surface, using Vickers method, and with Hanemann’s micro-hardness tester (Carl Zeiss, Jena, Germany), mounted on Neophot 2 microscopy with a load of 1 N. As part of the experiment, measurements of the roughness of coatings created with plasma spraying were taken with Olympus LEXT OLS 4100 confocal microscope (Nikon Instruments, Tokyo, Japan).

For testing the thermal and physical properties of the ductile cast iron and the WC–Co + Cr/ductile cast iron type system, a dilatometric analysis was performed—the values of materials’ ΔL/L dimension changes, and thermal expansion coefficients in solid state and in the function of temperature were defined; temperature conduction was measured with the LFA method. The samples of ductile cast iron and ceramic coating with dimensions of ϕ 3 × 30 mm were subject to thermal expansion measurements with the use of a high-temperature NETZSCH DIL 402C/4/G dilatometer (Erich NETZSCH GmbH&Co. Holding KE, Selb, Germany) in argon inert atmosphere in the temperature range of 100–700 °C, with a heating-up velocity of 5 K/min. To perform measurements of thermal conduction, Netsch LFA 427/4/G laser-flash type device was used, utilising the pulse technique to determine the temperature conduction coefficient (a) in the temperature range of 20–700 °C, i.e., the laser-flash method, consisting in heating up the front surface of square-shaped sample with a short laser pulse, which results in a temperature increase in the sample at its opposite surface, measured in the function of time with the use of an IR detector. The measured signal makes it possible to determine temperature conduction a and calculate thermal conduction λ based on the relation of:λ(T) = a(T)cp(T)d(T)(1)
where cp—proportionality factor of specific heat.

Measurement of internal stresses in the sprayed coatings was conducted with a non-destructive X-ray method (so-called sin^2^y). The research was conducted with an X-ray diffractometer (Brucker, Billerica, MA, USA), using monochromatic radiation of a cobalt anode lamp. The measurement of stresses was performed on a flat surface of a sample in 4 points. Both, the determination of measurement parameters and finding the location of diffraction lines with the assumed y angles was executed based on company APD or XRD Commander software (Bruker AXSLLC, Madison, WI, USA) for phase analysis, which is an equipment of the used research apparatus. The obtained experimental interplanar distances of *d_hkl_* and X-ray flexible constants for the researched material were input data for the software calculating internal stresses’ values. For measurements of internal stresses in the carbide coating, the following were adopted: reflection (211) and adopted elastic constants, i.e., Young’s module, 241 GPa, and Poisson factor, 0.23.

### 2.3. Mechanical and Tribological Testing

The strength of the coating/substrate bond was determined during a 3-point bend test on an INSTRON 8800 M machine (Instron Norwood, MA, USA), using a specially designed holder for coating/substrate type samples, with dimensions of 100 mm × 15 mm × 5 mm. The spacing of supports was 70 mm, and the deformation rate—1 mm/min. For one-time testing, 3 samples were used. Surfaces of fractures were observed after the 3-point bending strength test with the use of the scanning electron microscope (FEI Scios FEG firmy Thermo Fisher Scientific, Waltham, MA, USA).

The scratch bond strength tests on the coating were carried out using a multifunction measuring platform (Micro-Combi Tester, Buchs, Switzerland) equipped with Anton Paar scratch test heads according to the standard [23,24]. The tests were carried out on the cross-sectioned samples embedded in resin and then polished in a standard way as metallographic samples. The scratch test is done under constant load and the indenter moves from the substrate, through the coating, and into the resin where the sample is embedded. The following test parameters were used to produce scratch on each specimens: indenter (stylus) type Rockwell diamond; stylus radius 100 µm; constant normal loads of 10, 20 and 25 N; scratch length 1.1 mm and scratching speed of 1.2 mm/min. The scratch bond strength was evaluated using well-established formulation reported by other researches [24]. After the test, the geometric values of the resulting one-shaped fracture was also measured: cone length Lx, with Ly and cone angle (image of the cone fracture area was taken immediately after scratching using on light microscope). Detailed microstructural investigations of the cone fracture area were performed with a SEM (FEI Scios FEG firmy Thermo Fisher Scientific, Waltham, MA, USA).

Abrasion resistance testing in an abrasive suspension of ductile cast iron samples and WC–Co + Cr/ductile cast iron coating system was conducted with the use of a device for testing abrasive wear resistance of coatings and structural materials. An abrasibility test was performed in a water suspension of Cr_3_C_2_ (with a mean grain size of below 0.1 mm) in ambient temperature, and with the following parameters: test time 3600 s, rotation speed 300 L/min, applied load 50 N. The abrasion centre consisted in a still plate made of the tested material and a steel sphere rotating with a pre-set speed. The plate was pushed against the sphere with the pre-set force. The abrasive suspension was supplied to the abrasive contact sphere. The test process was recorded with the use of a computer and specialist software. The software made it possible, on an ongoing basis, to calculate the following: load, rotational speed, temperature of the friction pair, abrasion depth and value of wear velocity of both friction elements. The worn surface of the WC–Co + Cr deposited on ductile iron was characterized by scanning electron microscopy (FEI Scios FEG firmy Thermo Fisher Scientific, Waltham, MA, USA).

## 3. Results and Discussion

### 3.1. Microstructure

As a result of high-velocity spraying, dense WC–Co + Cr composite coatings were deposited on the ductile cast iron substrate. The results of microstructure observations of the obtained coatings carried out with a light microscope are shown in Figure 1. These coatings are characterised by a compact structure with a small number of visible pores, without cracks and with good adhesion to the substrate (the interface between the substrate and coating is continuous), which points to favourable conditions of the spraying process, providing for proper adhesion of a coating to a substrate. In the microstructure of cast iron, next to the coating/substrate interface, from the substrate side, there were no changes observed after the spraying process (the cast iron matrix, both initially and after spraying, was ferrite and pearlite—Figure 1d). During the spraying process, soft metals, acting as a matrix, are subject to strong deformation, while ceramic particles remain non-deformed, acting as particles reinforcing composite coating. Both high velocity and high temperature that accompany powder particles transferred in the direction of the substrate favour the formation of a ‘splat by splat’ coating, with good adhesion between one another and mechanical bonding with the substrate’s material [8,15]. Moreover, the combination of high temperature and velocity should simplify the deformation of the powder mixtures upon impact, so solidifying splats could adapt to the surface of the previously deposited layer and fill pores and defects. It is worth stressing that the sprayed powder grains form lamellas, which are deformed only to a slight degree. This is a result of the influence of a considerably lower temperature of the gas flame (kerosene and oxygen), and a short time of powder grains’ occupancy of high-velocity spray jet [4].

The results of micro-hardness measurements revealed a high hardness of HVOF sprayed composite coatings. The mean micro-hardness of a WC–Co + Cr composite coating is 2523HV0.1 ± 30. The high porosity of coatings is a result of a deformation hardening effect (owing to the high acceleration of powder particles towards the substrate during the deposition process), as well as the presence of very hard, super-fine ceramic particles. Following thermal spraying of the composite coating, almost an 11-fold increase in hardness of the ductile cast iron is observed, as compared with the initial condition, i.e., without coating (230HV0.1 ± 30). In the bonding area of the coating and substrate, a high degree of substrate deformation is visible, caused by depositing powder particles that form the coating, which greatly improves the bonding between the coating and substrate.

The mean porosity of the WC–Co + Cr composite coating is 4.1% ± 0.2. A relatively low porosity value is due to the high impact velocity of the coating particles, which causes high density and high cohesive strength of individual splats [4,7]. It seems that the admixture of Cr particles favourably conditions the reduction in coating porosity through the formation of a solid solution from Co, effectively binding carbide grains. It is also worth stressing that the composite coating is characterised by surface roughness, and the roughness parameter value Ra is 5.36 ± 0.6 μm (Figure 2). It is generally believed that the roughness of the coating decreases with increasing speed of the sprayed particles, and the improvement of the degree of melting of the particles [25]. The results of many studies indicate that lower surface Ra values (below 4 μm) of coating sprayed were obtained using optimizing three process parameters of HVOF including flow rates of kerosene and oxygen as well as spray distance.

From the SEM micrograph shown in Figure 3, bright carbide grains can be clearly seen in the WC–Co + Cr composite coating, which are embedded in a dark cobalt-chrome matrix. The conducted EDS analysis has shown different coating composition, depending on the researched micro-area. Light grains in the WC–Co + Cr composite coating are a phase with a high tungsten content (the EDS spectrum of point 1), which means that they are tungsten carbide grains, and, on the other hand, the dark matrix is an area rich in cobalt with a small content of chromium (the EDS spectrum of point 3). The coating has no cracks nor partially molten particles. Moreover, following the spraying process, a considerable fine fragmentation of tungsten carbide grains in the coating, and in the range of 125–500 nm, as well as the predominant amount of spherical tungsten carbide particles, which indicates that WC carbide grains are ‘molten in’ during spraying in the bonding phase—Co and Cr alloy.

The phase composition analysis, conducted with the XRD method (Figure 4), indicated that, within the coating composition, the following phases may be distinguished: WC, W_2_C, and Cr. The mean size of crystallites, calculated with the use of Williamson–Hall analysis for the WC phase, amounts to 50 nm, for the W_2_C phase—33 nm, and for the Cr phase—10 nm. It is worth stressing that the values of the mean crystallite sizes of individual phases provide evidence for the nanocrystalline character of the coating. Based on X-ray testing of WC–Co + Cr coatings, volume fractions of individual phases were also determined. The content of the WC phase amounted to 78.5%, and the content of W_2_C and Cr phases amounted to 6.6% and 14.8%, respectively. The coating features a relatively low content of W_2_C (which is a product of WC decomposition as a result of spray influence onto WC powder grains), which points to a low degree of decomposition of WC carbide to W_2_C. During the spraying process, WC dissolves in the binder, C oxidises to CO or CO_2_, and, during cooling, W precipitates as W_2_C, but the range of this reaction is limited in the case of the WC–Co + Cr composite coating. When determining parameters of the plasma spraying process, one of the aims should be to minimise the degree of decarbonisation of carbide particles, which guarantees a high hardness of a coating and its wear resistance [9,10].

It is worth mentioning that the WC phase (2200HV50) is characterized by a lower, hardness as compared with the W_2_C phase (3000HV50), but also a 2-fold higher elasticity module value (695 GPa), as compared with W_2_C (415 GPa) [26]. Thus, the formation of such a multi-phase coating structure, i.e., with a higher content of the retained WC and a limited number of brittle phases, should not have a detrimental effect on anti-wear properties. There were no peaks coming from Co detected (only C), and this indicates that the matrix may have an amorphous or nanocrystalline structure. This may be due to the very high cooling rate in the HVOF method spraying process. At the same time, it should be stressed that in the coating there are WC peaks with high intensity at 2θ, and in the range of 32–48°, and the intensity of W_2_C and Cr peaks is reduced. This points to the fact that in the thermal spraying process, an amorphous region emerged, which resulted from the dissolution of carbide or chromium and/or diffusion in the matrix, which leads to supersaturation of the matrix [11]. The formation of amorphous phases is caused by the combination of the extremely high cooling rate of melted powder droplets and the kinetics of slow crystallization in the coating spraying process [8]. It should be stressed that hardening of the binder phase, as a result of the creation of amorphous or nanocrystalline phase, leads to a high increase in coating hardness, but it reduces fracture toughness [27]. According to some authors, the presence of an amorphous structure and the presence of carbides provide for better cohesion between hard WC particles and the binder phase, as well as improve, both, ductility, and wear resistance.

Detailed microstructural testing of the coating, conducted on a thin TEM film from the cross-section of the sample, evidenced a nanocrystalline structure with band characteristics. In the coating’s microstructure, there are longitudinal bands with a thickness of 150–400 nm, aligned in parallel against each other (Figure 5), inside of which there are nanocrystalline carbide grains in the amorphous matrix. Amorphism of the matrix is confirmed by a diffractogram, which shows only halo-rings. On the other hand, halo-rings being properties of amorphous structures and reflections characteristic for crystallites suggest that the coating features intermediate character between a crystalline and amorphous structure. The EDS (energy-dispersive X-ray spectroscopy) technique was used to obtain a chemical point composition analysis in the coating and to identify particles forming the coating: W, Co, and Cr. Detailed TEM observations also indicate large microstructural defects of the produced coatings, and this confirms the high deformation of the cobalt-chromium matrix and generation of compressive stresses.

Temperature relation curves of linear thermal expansion coefficient (Alpha) and mean linear expansion factor (T.Alpha) for ductile cast iron and the WC–Co + Cr/ductile cast iron coating system are presented in Figure 6a,b. For the coating system in the entire tested temperature range, the obtained coefficient values were lower than for ductile cast iron, which proves greater thermal stability of the WC–Co + Cr/ductile cast iron system. The course of changes in the temperature coefficient **a** in the function of temperature and the value of thermal conductivity λ are presented in Figure 7a,b. Based on the measurements taken, it has been evidenced that for the coating system, over the entire tested temperature range, the temperature conductivity, as well as thermal conductivity calculated on their basis, are greater than for the ductile cast iron. An analysis of the numerical values of relations a(T) and λ(T), set up in Table 3 and shown in charts, makes it possible to conclude that the coating system is characterised by improved temperature and thermal conductivity (up to dozen or so %) in the temperature range of 300–700 °C, as compared with the ductile cast iron. Thus, it may be presumed that the coating system will remove heat well, and quickly, during the wear process. However, it is worth noticing that there is a tendency to reduce thermal conductivity with the increasing refinement of tungsten carbide grains, but secondary carbide phases produced in the spraying process (detected with the XRD method) are responsible for the relatively high thermal conductivity of the tested coating system [9]. Moreover, due to the difference in thermal expansion factors of the coating and substrate in the spraying process, thermal stresses are generated; thus, the stress state is influenced by the calculated differences in thermal conductivity of the coating and substrate.

### 3.2. Mechanical Properties

In Figure 8, there is a comparison presented of results for the bending strength test of the WC–Co + Cr/ductile cast iron coating system and ductile cast iron in relation to the bending stress–deflection. The maximum values of bending stresses for the WC–Co + Cr/ductile cast iron coating system and ductile cast iron are 570 MPa ± 12 and 590 MPa ± 10, respectively. In the tested coating system and the ductile cast iron, bending curves have parabolic characteristics. Tough, with the ductile cast iron, there is a long range of deflection path in the bending curve, during which the stress rises gently, and then drops. The value of the deflection, following which there is a drop in stress leading to the failure of the sample, is 6 mm. On the other hand, for the coating system, there is no such a long range of deflection path. By comparing the obtained curves, one may state that, for the WC–Co + Cr/ductile cast iron system, there is a reduction in strength parameters of the bending process, and the deflection, is shortened to a value of 2 mm. It is worth pointing out that the WC–Co + Cr composite coating on the ductile cast iron limits dissipation of plastic deformation energy, and intensely rising stress causes crack propagation and a smaller deflection range. Solution strengthening of the cobalt matrix seems to be the key factor influencing the mechanical properties of WC–Co coatings [4,7,11].

It must also be stressed that an important factor influencing the reduction in the final strength of the coating system is constituted by internal stresses generated by differences in values of linear expansion factors for the coating and substrate. For the material the substrate is made of, the value of the linear expansion factor amounts to approx. 13.2·10^−6^ K^−1^, and for the WC-Co type coating—approx. 5.5·10^−6^ K^−1^ [12]. However, due to the value of the linear expansion coefficient Cr (6.2·10^−6^ K^−1^), the addition of Cr particles to the starting powder has a marginal effect on the reducing internal stresses. Observations of the fracture of the samples after the bending strength test, conducted with the scanning electron microscope (Figure 9), indicate that, in the WC–Co + Cr/ductile cast iron system, the destruction occurs along the coating/substrate interface.

Results of the calculated major stress tests (σ_1_, σ_2_), generated in the WC–Co + Cr composite coating and the orientation of the major stress σ_1_ of the tested sample, are all set up in the Table 4. In the tested WC–Co + Cr coating, sprayed on the ductile cast iron substrate, there are compressive stresses. There is a high degree of microstructural defects in the deposited coatings after the HVOF spraying process, and this confirms high deformations of the material, generation of compressive stresses, and their high concentration in the area of the coating/substrate interface. Due to the considerable difference of thermal and physical, as well as mechanical properties of the substrate and coating, generation of internal stress conditions occurs in the coating/substrate system. The size, as well as the distribution of these stresses, influence the mechanical durability of the coating–substrate bonding area, especially the coating adhesion to the substrate and its scratch bond strength [26]. Residual compressive stresses, generated at the coating/substrate interface, improve coating adhesion and play an important role in thermally sprayed coatings, providing for the improvement in the fatigue range (hinder fracture generation) and wear resistance [27].

### 3.3. Wear Resistance of Coating

Results of the scratch test, performed on the cross-section of coating/substrate type systems, with a constant load of 10, 20, and 25 N, are provided in the Table 5. For the WC–Co + Cr/ductile cast iron system, the calculated Acn values (projected cone area) increased with the load increase, which, ultimately, indicates that the system is characterised by good scratch bond strength in the tested load range.

A cone-shaped fracture occurs inside the coating, which points to cohesive destruction in the coating/substrate system (Figure 10). With maximum load (25 N), there are larger cracks in the WC–Co + Cr coating around the scratch, leading even to delamination between the coating and the substrate (adhesive destruction).

The introduction of Cr particles to the coating material influences the increase in the reinforcement of the cobalt matrix and scratch bond strength. The high hardness of the coating, with a relatively low Young’s module value and with a small number of brittle phases, formed during the HVOF method spraying, reduces the effect of coating and substrate deformation during scratching.

Due to the fact that composite coatings may be employed in various tribological systems as coatings of high scratch bond strength, the abrasion resistance tests were performed in conditions of wet suspension for ductile cast iron and the WC–Co + Cr/ductile cast iron coating system. The measurement results, presented in Figure 11, indicate that the WC–Co + Cr/ductile cast iron coating system features, evidently, lesser wear, as compared with ductile cast iron (through the depth of wear). Moreover, a greater erosion intensity, in the case of the substrate deposited with the coating, has been confirmed. The groove for the coating system is 22% smaller than for the substrate not deposited with the coating.

Observations of the surface topography of the WC–Co + Cr/ductile cast iron type samples after the erosive abrasion test (Figure 12) indicated that the coating wear mechanism mainly consists in the removal of tungsten carbide particles through spalling, cracking and pulling out WC grains from the metallic matrix—Co–Cr alloy—as a result of repeated impacts of Cr_2_C_3_ grains in the erosive suspension.

On the eroded surface, lips, craters, cracks, and micropores may be observed. It is worth mentioning that the wear surface is relatively smooth, but microcuttings are present in the area of the cobalt-chromium matrix, which, in consequence, leads to troubles in exposing carbide particles and their removal. Elements, such as cuttings and craters, surrounded by lips, point to the plastic wear properties, and, on the other hand, cracks and spalls—to the brittle character of wear [28]. Based on the above findings, it may be suggested that the mechanism responsible for the coating wear is complex. Moreover, WC particles may act as subsequent abrasive solids, as very brittle phases, and may favour accelerated crack propagation. In the location of carbide removal, a relatively soft Co–Cr matrix is subject to further action of eroding particles, which cause deformation and removal, creating empty spaces [29]. The surface is thus subject to microgrinding, microcavity, as well as microindentation. These deformation processes lead to a local strengthening of the material and, as a result, to exhaustion of plasticity reserve and hardening of the surface layer of the substrate. With the increase in the layer hardness, the critical coefficient of stress concentration K_IC_ decreases, which may cause a sudden drop in resistance to abrasive wear [30]. It is worth mentioning that a high volume fraction of hard WC particles in the composite coating, with its high fine fragmentation and good bonding with the cobalt-chromium matrix, provides for high wear resistance. In particular, high fine fragmentation of WC grains considerably reduces abrasive wear, and the thin layer, bonding WC grains of a spherical shape, makes it more difficult to remove WC grains from the matrix in the grinding process.

When analysing results of the conducted research, one may arrive at a conclusion that erosive wear of the coating depends not only on its mechanical properties but on morphological parameters of the coating’s microstructure, and, especially, the percentage of hard WC particles, their size, shape, and adhesive bonding strength between WC particles and the cobalt matrix, as well as stress extent causing their decohesion.

## 4. Conclusions

Based on the conducted research and results of the analysis, the following conclusions have been formulated:The composite coating (WC–Co + Cr), deposited with the HVOF method onto the ductile cast iron, is characterised by low porosity (about 4%), compact structure, good adherence to a substrate, and high hardness (maximum hardness is obtained as 2523 HV0.1). In the coating’s microstructure, there are finely fragmented WC particles embedded in a cobalt-chromium alloy matrix, reaching nanocrystalline sizes. TEM tests confirm properties of the band structure of the composite coating, consisting of nanocrystalline tungsten carbide grains embedded in an amorphous cobalt-chromium matrix.Composite structure of the WC–Co + Cr coating provides good fracture toughness. Destruction is visible along the interface of the coating/substrate type system. Cracks are initiated in the area of interface, between coating and substrate, and do not develop into a crack in the substrate material. The system with the composite coating is characterised by high scratch bond strength in the load range of 10–25 N. Over 25 N, the coating is subject to substrate delamination, and the destruction mechanism is adhesive.Residual stresses, generated at the coating/substrate interface, are of compressive character and are improving adhesions of the coating to the substrate.WC–Co + Cr composite coatings, generated with the HVOF method on the ductile cast iron substrate, have good resistance to erosive wear. The composite coating (WC–Co + Cr) has better (by almost 22%) resistance to abrasion, in an abrasive suspension, than the ductile cast iron. The surface morphology, after the wear resistance testing in an abrasive suspension, indicates that the wear mechanism is associated with the formation of craters, lips, micro-cuts in the cobalt-chromium matrix, and the cracks in the area of the pores and in the area of the boundary of the carbide-matrix junction. The adhesive and abrasive wear are determined on worn surface.

## Figures and Tables

**Figure 1 materials-14-03282-f001:**
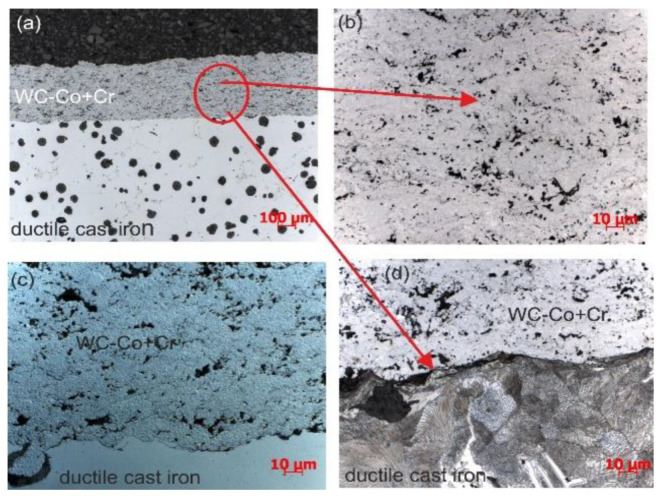
Microstructure of the composite coating (WC–Co + Cr) deposited on ductile cast iron: (**a**) LM image; (**b**) magnified area selected in Figure 1a; (**c**) details of the coating structure in differential interference contrast (DIC); (**d**) cast iron structure composed of ferrite and perlite.

**Figure 2 materials-14-03282-f002:**
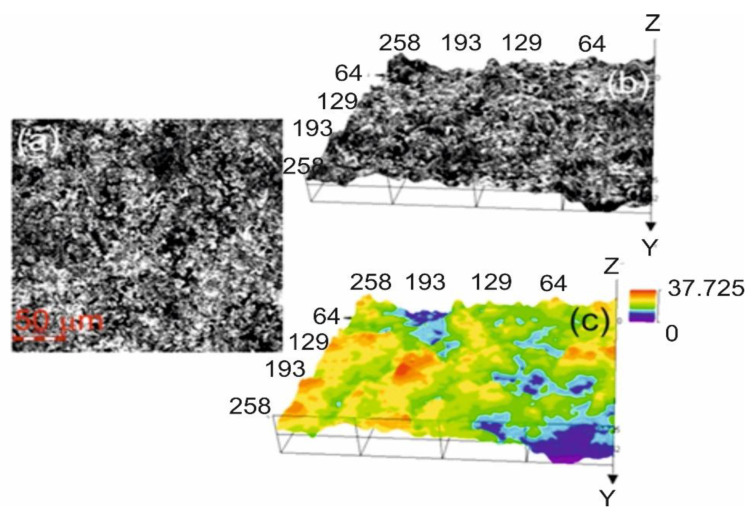
Surface texture of the (WC–Co + Cr) coating deposited on ductile cast iron created using a confocal microscopy; (**a**) 2D image; (**b**) 3D image; (**c**) 3D height mode.

**Figure 3 materials-14-03282-f003:**
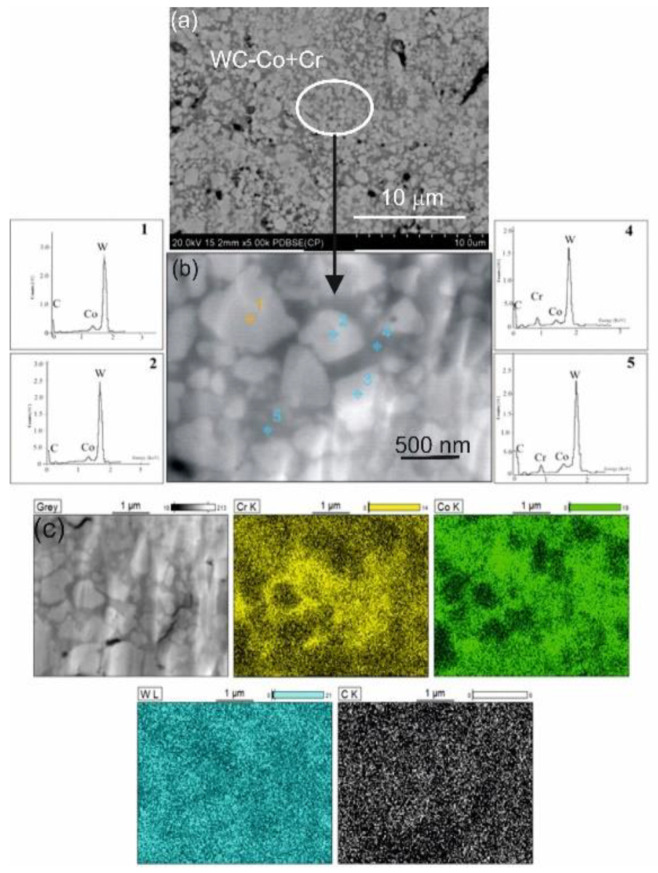
(**a**) Scanning micrographs of the composite coating (WC–Co + Cr) deposited on ductile cast iron interface with; (**b**) EDS spectra taken from the marked points: 1, 2, 4 and 5; (**c**) map of distribution of concentrations of Cr, Co, W, C taken from the region of interface.

**Figure 4 materials-14-03282-f004:**
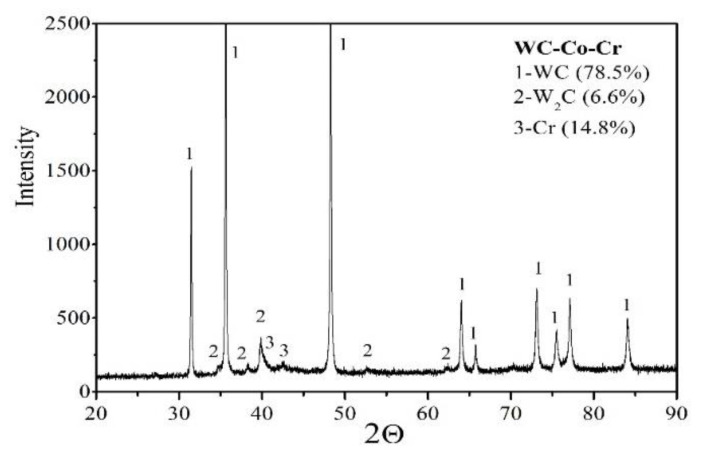
X-ray diffraction pattern of the composite coating (WC–Co + Cr) deposited on ductile cast iron by HVOF.

**Figure 5 materials-14-03282-f005:**
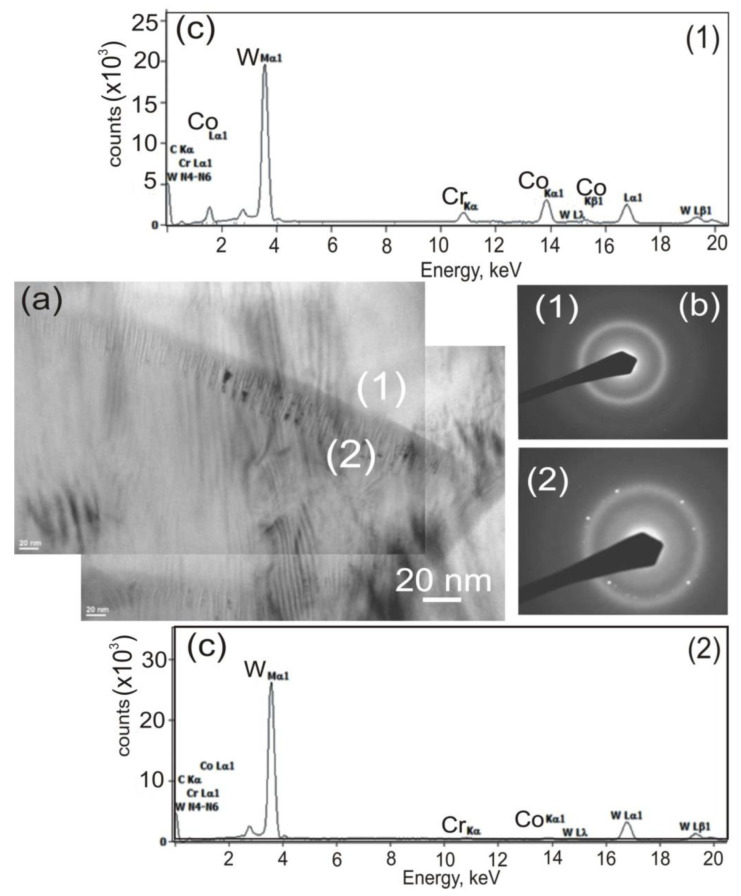
TEM analysis of the composite coating (WC–Co + Cr) deposited on ductile cast iron: (**a**) TEM image; with representative (**b**) area diffraction pattern indicates the formation of nanocrystalline structure; and with corresponding (**c**) EDS spectra taken from the marked points.

**Figure 6 materials-14-03282-f006:**
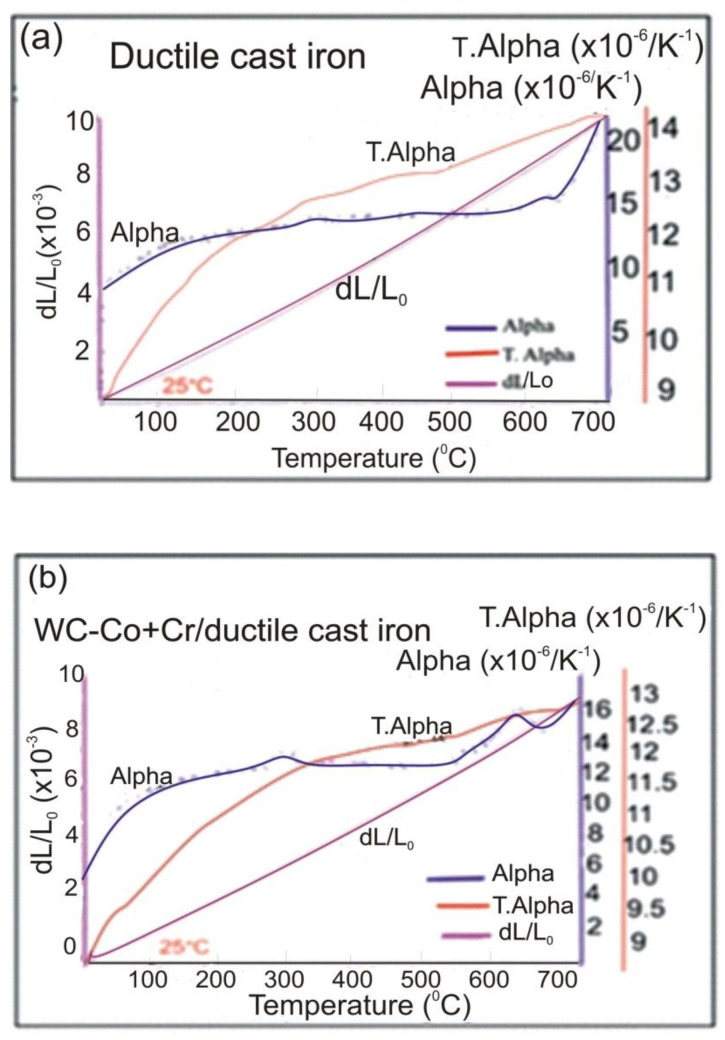
Plotted relationships between the thermal expansion (dL/Lo), the linear thermal expansion coefficient (Alpha), the average thermal expansion coefficient (T.Alpha) and temperature for: (**a**) ductile cast iron and; (**b**) WC–Co + Cr/ductile cast iron.

**Figure 7 materials-14-03282-f007:**
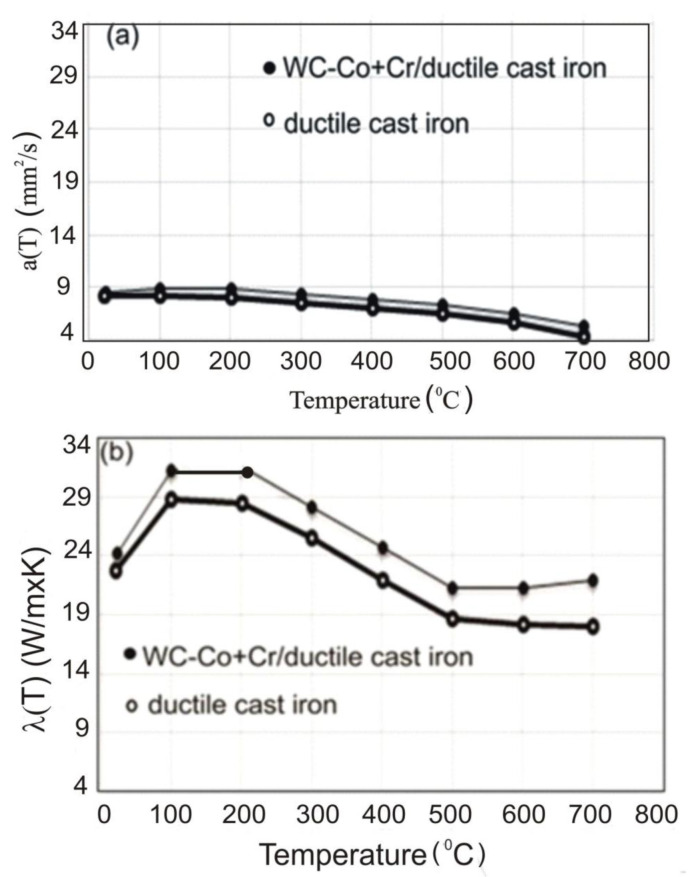
(**a**) Plotted relationship between temperature conductivity a(T); and (**b**) thermal conductivity λ(T) for: WC–Co + Cr/ductile cast iron and ductile cast iron.

**Figure 8 materials-14-03282-f008:**
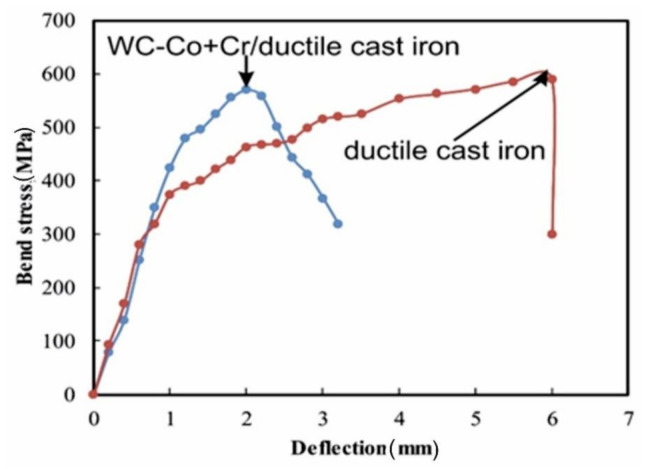
Bend test curves recorded for the ductile cast iron and system type: WC–Co + Cr/ductile cast iron.

**Figure 9 materials-14-03282-f009:**
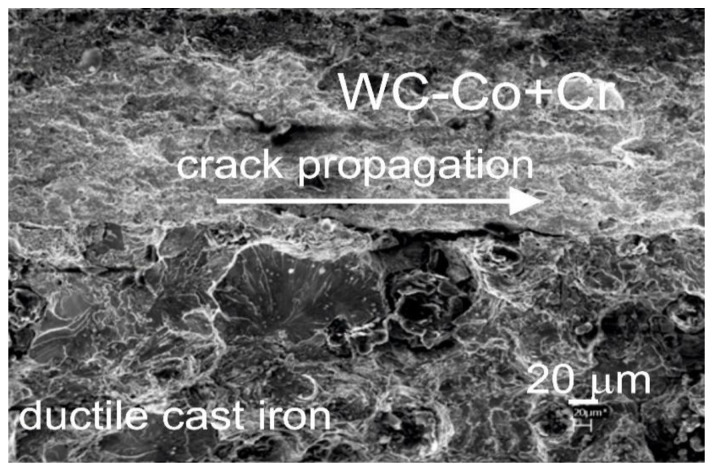
Scanning micrographs of the fracture surface of the WC–Co + Cr/ductile cast iron system after bend test.

**Figure 10 materials-14-03282-f010:**
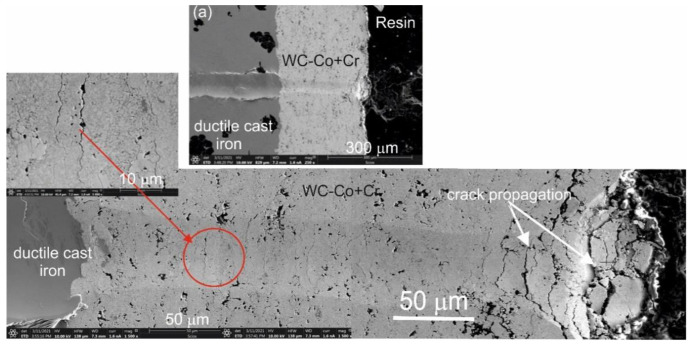
SEM micrograph of the cross-sectional area of the WC–Co + Cr/ductile cast iron system cracked after the scratch test at load: (**a**) 10 N; (**b**) 20 N and (**c**) 25 N.

**Figure 11 materials-14-03282-f011:**
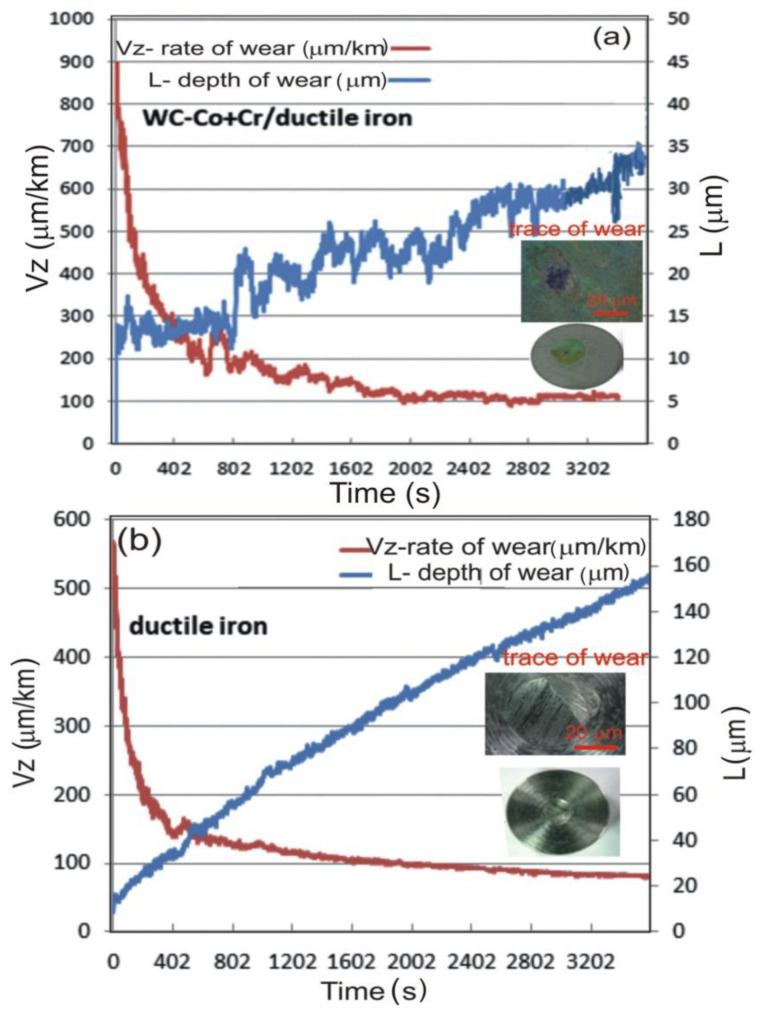
The results of the wear resistance tests of: (**a**) (WC–Co + Cr/ductile cast; (**b**) ductile cast iron.

**Figure 12 materials-14-03282-f012:**
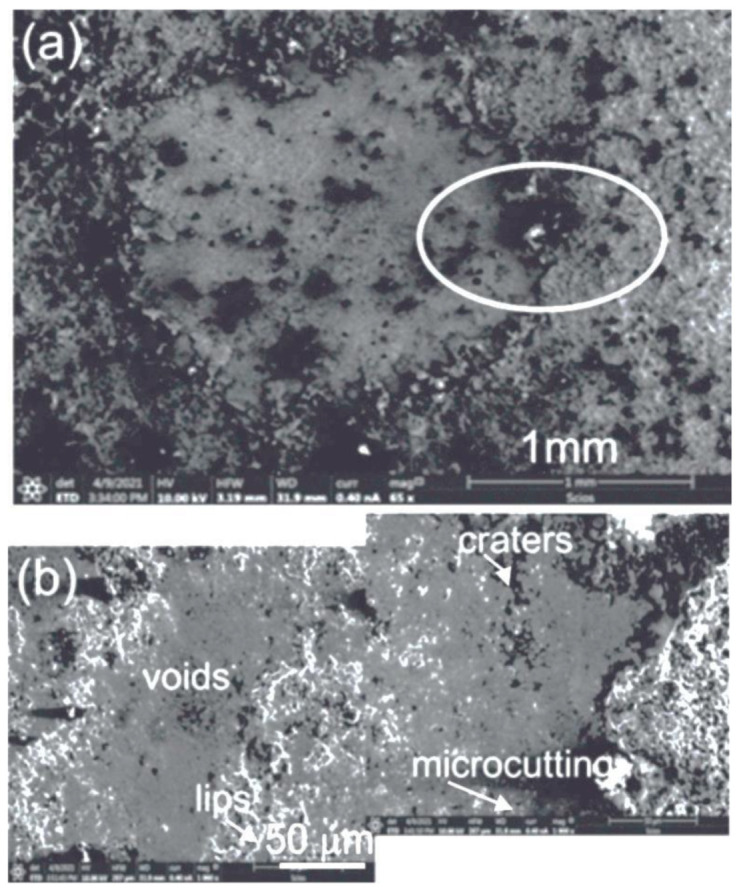
(**a**) SEM micrograph of the wear area after the erosion test of the composite coating (WC–Co + Cr); (**b**) magnified area selected in Figure 12a.

**Table 1 materials-14-03282-t001:** Mechanical properties of EN-GSJ-500-7.

Tensile Strength (MPa)	Conventional Yield Point (MPa)	Elongation (%)	Hardness (HV)	Elastic Modulus (GPa)
500	340	7	230	169

**Table 2 materials-14-03282-t002:** HVOF spraying parameters.

Gun Movement Speed (mm/s)	Oxygen (L/min)	Kerosene (L/h)	Powder Feed Rate (g/min)	Powder Feed Gas (L/min)	Spray Distance (mm)
583	944	25.5	92	Nitrogen, 9.5	370

**Table 3 materials-14-03282-t003:** Temperature and thermal conductivity coefficients in function of temperature for ductile cast iron and system: (WC–Co + Cr)/ductile cast iron.

Temperature (°C)	100	200	300	400	500	600	700
Substrate	Ductile Cast Iron						
Temperature conductivity, mm^2^/s	8.13	7.96	7.58	7.06	6.42	5.61	4.32
Thermal conductivity, W/mK	28.88	28.45	25.59	21.96	18.66	18.22	18.08
Coating	WC–Co+Cr						
Temperature conductivity, mm^2^/s	8.82	8.82	8.36	7.93	7.32	6.57	5.24
Thermal conductivity, W/mK	31.33	28.25	28.20	24.66	21.28	21.35	21.94

**Table 4 materials-14-03282-t004:** Results of an analysis of residual stresses in WC–Co + Cr coating.

Description	WC–Co + Cr
Internal stress σ_1_ (MPa)	−130 ± 60
Internal stress σ_2_ (MPa)	−340 ± 80
Orientation of the main stress σ_1_ (clockwise from the direction marked on the sample) 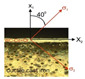	40° ± 30°

**Table 5 materials-14-03282-t005:** Averaged scratch bond test results of WC–Co+Cr coating.

Coating /Load	10 N			20 N			25 N		
–	Lx (µm)	Ly (µm)	A_cn_ × 10^−3^ (mm^2^)	Lx (µm)	Ly (µm)	A_cn_ ×10^−3^ (mm^2^)	Lx (µm)	Ly (µm)	A_cn_ × 10^−3^ (mm^2^)
WC–Co + Cr	57.92	76.19	4.41	20.48	185.16	22.31	75.85	90.12	51.02
Cone shape fracture	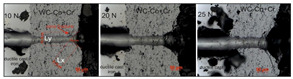								

## Data Availability

Data is contained within the article.

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
