# Peer review of "Investigation on Microstructure, Mechanical and Wear Properties of HVOF Sprayed Composite Coatings (WC–Co + CR) On Ductile Cast Iron"

_materials, 2021, doi:10.3390/ma14123282_

Round 1

Reviewer 1 Report

  1. The current study investigates the effect of High Velocity Oxygen Fuel composite coatings made from (WC-Co+Cr) which are applied on ductile cast iron. For this the authors study the microstructural condition and the wear and mechanical performance such as scratch test and bending strength, cracks, microhardness and delamination. The authors found that authors found that adding of chromium particles improved the wear resistance of the material.
  2. Please consider reviewing the abstract and highlight the novelty, major findings and conclusions. Especially highlighting the main findings.
  3. please add the full meaning of HVOF where it first appears in the abstract (i.e. line 12 and not line 15)
  4. the literature review must be improved and extended. The authors must discuss on past studies similar to this work and mention what they did, what was their main findings and explain how does the current work brings new knowledge and difference to the field.
  5. After line 80 the authors should attempt to answer the following question: What is the research gap did you find from the previous researchers in your field? Mention it properly. It will improve the strength of the article.
  6. The authors should add tables showing the mechanical properties of the steel used and the coating applied.
  7. The materials and methods section should be improved by adding images and figures of the equipment, tools and materials used in this study as it is an experimental work care should be given to all details of experimental analysis. Add pictures of coated samples, samples before and after testing, machines used…etc
  8. Line 216-217 please support this claim by a reference, also what about previous studies what did they found when they applied this or similar coatings on steel materials? Please provide more in depth discussion everytime you make a claim or report a finding.
  9. Line 242 What is the hardness of the steel material without the coating? Was the difference significant
  10. Ok now I see you mentioned it in line 247, but please mention specifically how much difference was there in % terms for example say it was 10% higher or lower..etc
  11. Line 256 the authors did not mention they measured the surface roughness in the abstract? Please check this
  12. Line 254 please support this with a reference and expand further by discussing and comparing this with past literature, for example what did they find in terms of Ra was it similar, higher lower..etc please provide more detailed discussion each time you report a finding or make a claim
  13. Extensive editing of English language and style required
  14. Figures 5 and 6 have poor resolution, please replace them with ones with better quality
  15. The paper needs proper formatting, there are variations in formatting everywhere especially after page 11
  16. Line 450 “thermal deformation” the definition you mentioned here in incorrect please check this carefully and update
  17. Thermal deformation: Thermal deformation is the property of a substance to expand with heat and contract with cold, customarily called temperature deformation. It is expressed by linear expansion coefficient α.
  18. Line 504-506 this is a very generic claim which needs further evidence to support it, please check this sentence, revise and improve it.
  19. Figures 10 have poor resolution, please replace them with ones with better quality
  20. The results are merely described and is limited to comparing the experimental observation. The authors are encouraged to include more discussion and critically discuss the observations from this investigation with existing literature.

Author Response

The authors would like to thank warmly for the valuable comments included in the review of the article. These remarks made it possible, first of all, to organize the terminology used in the work and a thorough analysis of the results in relation to literature reports.

The responses to all the comments included in the review are presented below:

  1. The summary introduces content summarizing the significant results of experimental research
  2. Given the full meaning of the abbreviation HVOF in the previous line.
  3. Literature review was carried out - the list was extended to include the applicant's position relevant content to the issue presented.
  4. Content emphasizing the concept of modifying the composition has been introduced chemical input powder through the introduction of particles metallic to ceramic powder.
  5. A table with the mechanical properties of cast iron has been introduced (table 1).
  6. The equipment used for testing is standard (manufacturers are given). Including photos would increase the volume considerably article. Also, photos of the samples used for the tests (and after   research) would not extend the substantive essence of the experiment.
  7. The statement (lines 216-217) was supported by a literature reference:

9 and 10. The article gives an 11-fold increase in the hardness of the substrate after spraying  coatings. It seems that such a record is sufficient .

  1. The results of the roughness measurement using a confocal microscope (Fig.2) are shown graphically: (2D image, 3D image and 3D height mode). In the experimental part, information about the roughness test is given.
  2. The results of Ra measurement are discussed in more detail (line 256).
  3. Editing of English and style was carried out.
  4. The quality of Figures 5 and 6 has been improved.
  5. Corrected formatting of the article.
  6. and 17. Revised term - thermal deformation (line 450).
  7. Part of the text contained in lines 504-506 has been corrected.
  8. The quality of Fig. 10 has been improved.

Reviewer 2 Report

This manuscript provides a systematic characterization of HVOF sprayed composite coatings. The introduction includes a vast range of literature to justify the need for this research. The experimental section is clear and systematic. The coating was characterized by X-ray diffraction (XRD), scanning electron microscope (SEM), transmission electron microscope (TEM), and energy-dispersive X-ray spectroscopy (EDS). Additionally, bending tests and hardness tests were conducted. It was concluded that a significant improvement occurs when Cr particles are added. The conclusion was based on the experimental observation and the manuscript will attract readers. However, the following needs to be addressed before acceptance:

  1. Please avoid grouped citation such as [1-3], [4-6], [11-13] and, [15–17].
  2. Please avoid different naming conventions. For example, in some places, the materials are mentioned as WC12Co, and in some places, the materials are mentioned as WC-12Co.
  3. Page 2 line 92: Please make necessary corrections for "grain size of 45+5 μm".
  4. Page 3 line 99: Please write down the meaning of Rm, Rp0.2, and A5.
  5.  Please use the same type of hardness measurement throughout the paper. For example, on page 3 the hardness was reported on the scale of Brinell hardness. On the other hand, on page 5 line 242 the hardness was reported on the Vicker scale.
  6. Page 3 line 256: It would be good to report the roughness profile in a figure showing all the textures such as lay direction, waviness spacing etc.
  7. Please use the same formation throughout the paper for fig. and figure. Please use either of them but do not use both.
  8.  Page 12 line 501: Is it Table 3?
  9. Please provide a clear direction of major stresses (Sigma1, sigma2) in a figure. The directions are not clear from table 3.
  10.  Please provide a hypothetical reason for the variation of thermal conduction and thermal conductivity depending on temperature. Why the thermal conduction is highest between 100-200 degrees in figure 6?
  11.  Please make necessary changes to figures 4, 8, and 11. they are confusing.

Author Response

I would like to warmly thank you for the valuable comments included in the review of the article. These remarks made it possible, first of all, to organize the terminology used in the work.

The responses to the comments included in the review are presented below:

  1. The allegation of improper grouping cited in the introduction - the literature citation has been corrected and ordered
  2. and 3. Uniform nomenclature was introduced and grain size designations in carbide powder were improved
  3. A tabular entry (Table 1) has been introduced concerning the mechanical properties of cast iron and verbal designations of these sizes have been given
  4. As suggested, the value of the hardness of cast iron on the HV scale was introduced (as for the materials used)
  5. The results of the roughness measurement using a confocal microscope are presented graphically (Fig. 2): (D image, 3 D image and 3D heigh mode)
  6. A uniform record of the submitted drawings has been introduced - Figure1, Figure 2, etc.
  7. and 9. The missing number of the table concerning the results of internal stresses was introduced and the directions of the main stresses on the tested sample were presented.
  8. Measurements of the temperature conductivity coefficient a were made in high vacuum, in the range from ambient temperature to 700oC. The calculations of the thermal conductivity coefficient were performed according to the Cape-Lehman mathematical model. Each measurement point (temperature value) consists of three measurements, ie 3 fired laser "shots" (pulses).

       The calculated values ​​of thermal conductivity l are burdened with an additional error, as it is not possible to determine the specific heat and density of samples consisting of cast iron and a thin coating layer in the appropriate proportion. For the materials tested, the specific heat and density depend primarily on the properties of ductile iron. Therefore, the specific heat and density of ductile iron were adopted for the calculation of the thermal conductivity of cast iron + coating.

       With the temperature increase l in the range of 200-500°C, it significantly decreases both for the coating system and for cast iron. However, in the range of 100-200°C, in the case of the coating system, l does not change, and for cast iron it decreases slightly. Perhaps the internal structure of the coating, the presence of the lamellae of the carbide phase, and thus the separation boundaries and the discontinuities formed in them, which should lower the values ​​of a and l, do not affect this scope. Due to the complex structure of the coating, it is difficult to understand the heat transfer mechanisms. The range from 200-700°C is important for the analysis, because in this range there may be changes in the internal structure (changes in the solid state). It is worth noting that above 500°C the dimensional stability of the system changes.

  1. Figures 4, 8 and 11 have been corrected as suggested, to make them more "legible"

Round 2

Reviewer 1 Report

The authors have improved the quality of the article signficantly, paper can be accepted

Reviewer 2 Report

The authors have addressed all the issues raised by the reviewer.

It can be publish now.